# Heavy Metal Pollution Characteristics and Source Analysis in the Dust Fall on Buildings of Different Heights

**DOI:** 10.3390/ijerph191811376

**Published:** 2022-09-09

**Authors:** Hanyang Song, Jinxiang Li, Lingjun Li, Jie Dong, Wenxing Hou, Ran Yang, Shanwen Zhang, Sida Zu, Pengfei Ma, Wenji Zhao

**Affiliations:** 1College of Resource Environment and Tourism, Capital Normal University, Beijing 100048, China; 2Beijing Municipal Ecological and Environmental Monitoring Center, Beijing 100048, China; 3Satellite Application Center for Ecology and Environment, Ministry of Ecological Environment, Beijing 100048, China

**Keywords:** dust fall, enrichment factor, heavy metal, source analysis

## Abstract

High-rise buildings block airflow, and dust accumulates on their upper surfaces. In this study, dust fall on the rooftops of low-, medium-, and high-rise buildings was sampled and analyzed to assess the degree of atmospheric heavy metal pollution. The Cr, Mn, Ni, Cu, Zn, As, Cd, and Pb mass fractions in dust samples were analyzed by microwave digestion/inductively coupled plasma-mass spectrometry. The average Cr, Ni, Cu, As, Cd, and Pb concentrations were highest on the rooftops of low-rise buildings, whereas those of Mn and Zn were highest on high-rise buildings. The cumulative indices for the eight heavy metals revealed a moderate pollution level for Zn on the rooftops of low- and high-rise buildings. Only the potential ecological risk index for Cd was very high, with a particularly high heavy metal-related ecological risk for low-rise buildings. The enrichment factor analysis and principal component analysis (PCA) demonstrated that Zn and Cd were strongly influenced by human activity. Zn, Cu, Cd, and Pb originated from traffic sources, Cr and Ni were derived from natural sources, and As was of industrial origin. The source analyses of rare earth elements were consistent with the heavy metal PCA results. In conclusion, our results provide a reference for hazard and source analysis of heavy metals in atmospheric dust fall on buildings of different heights.

## 1. Introduction

Atmospheric dust fall is particulate matter that naturally settles on the ground, representing a major urban pollution source [1]. Heavy metals in dust fall undergo accumulation and enrichment, eventually causing varying degrees of harm to humans [2]. Previous research on atmospheric dust fall focused mainly on source analyses [3], health risk assessments [4], and spatial distributions [5]. The research methods included the enrichment factor (*EF*) method [6], the index of geoaccumulation (*I*_geo_) [7,8], and principal component analysis (PCA) [9]. This study analyzed the concentration, enrichment, and sources of eight heavy metals in atmospheric dust fall in an urban area of Beijing. The heavy metal and rare earth element (REE) migration trends and the impact of human activity on the dust fall elemental composition were investigated to clarify the pollution characteristics and sources of heavy metals in dust fall on the rooftops of buildings with different heights.

According to the “Code for the Design of Civil Buildings” (GB50352-2019) [10], buildings with up to three floors and a height below 10 m are defined as low-rise buildings; buildings with four to nine floors and a 10–24 m height are considered medium-rise buildings; buildings with ten floors or more and a height of over 24 m are high-rise buildings. These buildings not only block the flow of air but their top surfaces also become the main place for dust accumulation, constituting an important secondary dust source.

Environmental protection standards are constantly improving as urbanization and industrialization relentlessly progress in China. Numerous highly polluting enterprises formerly in Beijing have relocated outside the city, improving the quality of the atmospheric environment. However, various industries and construction sites still operate in Beijing, contributing to the remaining environmental issues. Building density is steadily increasing in urban Beijing [11]. However, relatively little research has been conducted on dust fall reduction and re-entrainment on building rooftops. Hence, we examined the properties of atmospheric dust fall on buildings differing in height, the regional atmospheric pollution status, and meteorological conditions. In this manner, we determined the degrees of influence of various factors on the heavy metal composition of rooftop dust fall. The results of this work provide a scientific reference for predicting and controlling dust fall pollution on buildings in urban Beijing.

## 2. Methods

### 2.1. Study Area and Sample Collection

Beijing (116°20′ E, 32°03′ N) is the capital of the People’s Republic of China and the largest city in the northern part of the nation. It is also an important national political center and transportation hub. In the present study, 14 atmospheric dust fall sampling points were selected in typical urban areas of Beijing (Figure 1). Roads and residential areas surrounded the study area, and there was no industrial pollution source. The sampling points are evenly distributed, including five, four, and five on low-, medium-, and high-rise buildings, respectively. We attempted to ensure that adjacent sampling points avoided building height zoning of the same type to reduce the impact of randomness. They were set at the headquarters of Capital Normal University and divided into three categories according to building height. Sampling points 3 (4.3 m), 4 (8.5 m), 11 (6.7 m), 12 (4 m), and 14 (6.5 m) were located on low-rise buildings. Sampling points 1 (11.5 m), 7 (19 m), 8 (24 m), and 9 (14 m) were situated on medium-rise buildings. Sampling points 2 (38.8 m), 5 (41 m), 6 (30 m), 10 (45.5 m), and 13 (50 m) were placed on high-rise buildings. Dust fall was collected from the dust fall tank monthly between December 2020 and November 2021 to reduce the impact of extreme weather by increasing the sampling data time. To ensure sufficient dust fall collection, sampling would only be performed after at least one rain-free week to ensure sufficient dust fall accumulation. The collection was completed within one day. Sampling was conducted in strict compliance with the gravimetric method for the determination of ambient air dust fall (No. GB/T15265-1994) [12]. Each dust fall tank consisted of a glass cylinder with a flat and smooth bottom, and height and inner diameter of 30 ± 0.2 cm and 14 ± 0.17 cm, respectively. The dust fall tanks were fixed on hanging levers built in-house and set approximately 2 m above the roof surfaces. Each dust fall collection station was located on a rooftop, not sheltered from tall buildings, and remote from the road. Ethylene glycol was added to each dust fall tank to inhibit algal and microbial growth. The amount of ethylene glycol varied with seasonal temperature and humidity. Distilled water was also added to keep the bottom of the tank moist. The volume of water used to replenish the tank bottoms also varied with season. Before the end of sampling, plastic film was used to seal the dust fall cylinder so that its contents would not be lost during replacement and transport. After sample collection, rubber policeman tweezers were used to remove leaves and other foreign matter from the dust fall tanks. Any dust fall remaining on the tank walls was rinsed off with distilled water. A broom was used to sweep any dust fall adsorbed on the tanks’ walls into the collection liquid. In the laboratory, an empty evaporating dish and a porcelain crucible were dried in an oven at 105 °C for 3 h and cooled in a desiccator. Each vessel was weighed twice on an electronic balance (±0.01 mg; Sartorius AG, Göttingen, Germany) to 1/10^5^ precision. If the weight difference was ≥0.4 mg, the vessels were dried and weighed once again. To eliminate the influence of ethylene glycol, two aqueous ethylene glycol blanks were set while the dust fall samples were being evaporated. Evaporating dishes and porcelain crucibles containing the dust fall samples were dried and weighed as previously described. The net dry weight of the dust fall sample (in grams) was calculated as follows:(total weight of sample + evaporating dish + porcelain crucible) − (weight of empty evaporating dish + empty porcelain crucible).

### 2.2. Sample Treatment and Heavy Metal Detection and Analysis

An electronic balance with 1/10^5^ precision was used to weigh out 0.1-g samples. Each sample was then placed in a 50-mL PTFE digestion tank, to which 6 mL of HNO_3_, 2 mL of H_2_O_2_, and 0.25 mL of HF were added. The samples were microwaved in a MARS 1 digestion system (CEM Corp., Matthews, NC, USA). Each digested, cooled dust fall sample was transferred to a polyester bottle, and the liquid volume was made up to 100 mL with deionized water. The mass fractions of Cr, Mn, Ni, Cu, Zn, As, Cd, and Pb were determined by inductively coupled plasma-mass spectrometry (ICP-MS; Agilent 8800; Agilent Technologies, Santa Clara, CA, USA). ICP-MS has a low detection limit and can analyze multiple elements with high sensitivity. Yellow–brown soil was the reference material for soil composition analysis, and the national soil sample standard value GB07403 (GSS-3) was also used for quality monitoring. The relative standard deviation (SD) and determination error were controlled within ±3%. Two blank controls were set per sample batch. The average element concentrations in the blank samples were deducted from the gross measurements.

### 2.3. Enrichment Factor

The *EF* has been widely used to identify elemental pollution sources in the environment [13]. The *EF* is calculated as shown in Equation (1):(1)$$EF_{i}=\frac{{\left(C_{i}/C_{n}\right)}_{\mathrm{sample}}}{{\left(B_{i}/B_{n}\right)}_{\mathrm{background}}}$$
where *EF_i_* is the enrichment factor of element *i* measured in the atmospheric dust fall samples; *C_i_* and *C_n_* are the concentrations of the measured element *i* and the reference element *n*, respectively, in the sample; and *B_i_* and *B_n_* are the concentrations of the measured element *i* and the reference element *n*, respectively, in the background soil.

In *EF* analyses, Al, Ti, Fe, and Si are comparatively less influenced by human activity and have relatively stable chemical properties. Hence, they are usually selected as reference elements. According to the method of reference element selection reported by Fan Xiaoting et al., elements with natural sources, stable chemical properties, and high content in the crust were selected [14]. Here, Fe was selected as the reference element. Table 1 lists the evaluation criteria for the *EF*s.

### 2.4. Pollution Characteristics of Heavy Metals in Dust Fall

#### 2.4.1. Index of Geoaccumulation (*I*_geo_)

*I*_geo_ is a quantitative index determining the degree of heavy metal pollution in sediments [16,17,18]. It reflects the impacts of both anthropogenic activity and natural geological processes. In recent years, the geoaccumulation index method has been widely used to assess the extent of heavy metal contamination in atmospheric dust fall [19,20] and soils [21]. It has also been used in health risk assessments [22]. In the present study, *I*_geo_ was used to analyze the pollution characteristics of eight heavy metals in atmospheric dust fall and calculated as shown in Equation (2):(2)$$I_{\mathrm{geo}}=\log_{2}\left(\frac{C_{n}}{1.5B_{n}}\right)$$
where *I*_geo_ is the index of geoaccumulation, *C_n_* is the concentration of heavy metal element *n* in topsoil, and *B_n_* is the geochemical background of heavy metal element *n*.

Dust fall is transported mainly from surface soil under wind action. Surface soil represents a secondary environment affected by anthropogenic activity. Thus, secondary geochemical background values were obtained, and abnormal data were excluded [23]. The heavy metal pollution levels in the dust fall were determined based on the calculated *I*_geo_ values (Table 2).

#### 2.4.2. Potential Ecological Risk Index

The potential ecological risk index method evaluates the possible ecological hazards of heavy metals [24] based on their sedimentology, properties, and environmental behavior. The potential ecological risk index method introduces the toxicity coefficient $T_{r}^{i}$, which compares the relative toxicity levels of various heavy metals. The potential ecological hazard coefficient ($E_{r}^{i}$) for a single element and the potential hazard indices (*RI*s) of various heavy metals in atmospheric dust fall were calculated as shown in Equations (3) and (4) [25]:(3)$$E_{r}^{i}=T_{r}^{i}\times \frac{C^{i}}{C_{n}^{i}}$$
(4)$$RI=\overset{m}{\underset{i=1}{\sum }}E_{r}^{i}$$
where *C_i_* is the measured concentration of heavy metal element *i* in atmospheric dust, $C_{n}^{i}$ is the background value of heavy metal element *i*, and $T_{r}^{i}$ is the toxicity coefficient of heavy metal element *i*. $T_{r}^{i}$ reflects the toxicity level and biological sensitivity to heavy metal element *i*.

In the present study, the background values of the heavy metal elements in Beijing soil were selected. The potential ecological risk index method was used to evaluate the potential ecological hazard level of the dust fall on various buildings in Beijing. The toxicity coefficients of Cr, Mn, Ni, Cu, Zn, As, Cd, and Pb were 2, 1, 5, 5, 1, 10, 30, and 5, respectively [26]. The potential ecological risk assessment indices and their classifications are shown in Table 3.

### 2.5. Source Analysis of Heavy Metals

#### 2.5.1. Principal Component Analysis (PCA)

The sources of heavy metals in atmospheric dust fall are complex and extensive and are mainly divided into natural and human sources. In order to study the specific sources of dust fall on the top floors of buildings with different heights, PCA of eight heavy metals was carried out. PCA uses dimensionality reduction to convert multiple original environmental variables into a few comprehensive ones. It calculates the variance in these parameters and the characteristic quantities of the covariance matrix. In this manner, the predominant constituents of numerous environmental pollutants may be identified. As PCA concentrates and extracts environmental pollutant data, it is widely used to evaluate and analyze atmospheric dust fall pollution [27]. In this study, PCA was used to reduce the dimension of the dust fall data of eight heavy metal elements, understand the relationship between heavy metal elements in the dust fall, and further judge and identify the source contributions of heavy metals in the atmospheric dust fall so as to provide a scientific basis for understanding the atmospheric dust fall on buildings with different heights.

#### 2.5.2. Rare Earth Element Analysis

There are relatively few reports on the geochemical characteristics and sources of REEs in dust fall. Members of this group of elements have unique geochemical properties, and their compositions and distribution modes are unaffected by weathering, transportation, sedimentation, or diagenesis. Hence, they do not undergo obvious chemical separation in the environment. The content, distribution modes, and other parameters of REE in dust fall can serve as references for the sources of heavy metals in dust fall reduction [28].

## 3. Results

### 3.1. Content of Heavy Metal Elements in Dust Fall on the Rooftops of Urban Buildings

The concentrations of the eight heavy metals in the dust fall on the low-, medium-, and high-rise buildings are listed in Figure 2. The error range of the heavy metal content was obtained by adding and subtracting the SD from the average heavy metal content. The order of the heavy metal contents in the dust fall on low- and medium-rise buildings was Mn > Zn > Cr > Cu > Ni > Pb > As > Cd. The order of the heavy metal contents in the dust fall on high-rise buildings was Mn > Zn > Cr > Cu > Pb > Ni > As > Cd. The average Cr, Ni, Cu, As, Cd, and Pb concentrations were highest in the dust fall on low-rise buildings. The Mn and Zn concentrations were highest in the dust fall on high-rise buildings. Mn had the highest content in the dust fall and accounted for 44%, 51%, and 48% of the total heavy metal contents on low-, medium-, and high-rise buildings, respectively.

Table 4 shows the average heavy metal contents, SD, and coefficients of variation (CVs) in the dust fall on various buildings. The average Cr, Mn, and Zn concentrations were >100 mg·kg^−1^, the average Cu and Pb concentrations ranged from 1–100 mg·kg^−1^, and the average Cd concentration was <1 mg·kg^−1^ in the dust fall on all buildings. The average heavy metal content in the dust fall on medium-rise buildings was slightly lower than the background heavy metal content in Beijing soil. In contrast, the heavy metal content in the dust fall on low- and high-rise buildings exceeded the background heavy metal content in Beijing soil [29,30]. In the dust fall on low-, medium-, and high-rise buildings, the Cr, Mn, Ni, Cu, Zn, As, Cd, and Pb concentrations were 8.9-fold, 4.2-fold, and 4.4-fold; 1.1-fold, 0.99-fold, and 1.1-fold; 3.3-fold, 2.2-fold, and 1.9-fold; 4.8-fold, 3.8-fold, and 3.7-fold; 8.2-fold, 6.4-fold, and 9.5-fold; 2.0-fold, 1.6-fold, and 1.7-fold; 7.5-fold, 6.7-fold, and 6.7-fold; and 3.1-fold, 2.2-fold, and 2.5-fold higher, respectively, than the background contents in Beijing soil. Hence, anthropogenic activity, including urbanization, industrial emissions from the surrounding areas, and automobile exhausts, substantially contributed to dust fall in the study area.

### 3.2. Heavy Metal EFs

The *EF* of any element in atmospheric dust fall is near unity. Thus, the *EF* is affected mainly by natural sources. If *EF* > 5, then it is mainly affected by human activity [31]. Figure 3 compares the *EF*s in the dust fall on buildings differing in height. The average *EF*s were in the order of Zn (7.92) > Cd (7.36) > Cr (5.49) > Cu (4.11) > Pb (2.59) > Ni (2.48) > As (1.74) > Mn (1.07). The *EF*s were >5 for Zn and Cd in the dust fall on the buildings of all three heights. Table 1 shows that both Zn and Cd were significantly enriched by natural sources, especially anthropogenic activity. The *EF*s were 1–5 for Cu and Pb in the dust fall on all buildings. Thus, both Cu and Pb were moderately enriched. Furthermore, their enrichment was affected by natural sources and human activity. The *EF* was 1–2 for As in the dust fall on all buildings. Therefore, As was slightly enriched and predominantly affected by natural sources, rather than human activity. Cr was highly enriched in the dust fall on low-rise buildings, but moderately enriched in the dust fall on the medium- and high-rise buildings. Therefore, its pollution level was high. Ni was moderately enriched in the dust fall on low- and medium-rise buildings, but only slightly enriched in the dust fall on high-rise buildings. Its overall *EF* was in the range of 1–5. Thus, both natural sources and human activity influenced it to a certain degree. Mn was slightly enriched in the dust fall on low- and medium-rise buildings, but not in the dust fall on high-rise buildings. Hence, its pollution level was low, and it was somewhat affected by human activity. The *EF*s for Cr, Cu, Ni, Pb, Zn, As, Ni, and Cd decreased with increasing building height. The *EF*s for Cd and Mn reached their maxima in the dust fall on medium-rise buildings, whereas the *EF* for Zn reached its maximum in the dust fall on high-rise buildings. As a rule, building height had a certain impact on the concentrations of heavy metals in rooftop dust fall.

### 3.3. Pollution Characteristics of Heavy Metals in Dust Fall

#### 3.3.1. Geoaccumulation Index

Table 5 lists the geoaccumulation indices (*I*_geo_) for the heavy metals in the dust fall on buildings differing in height. The sum of all eight heavy metal pollution levels indicated that the pollution degrees for the buildings differing in height were as follows: low-rise (12) > high-rise (8) > medium-rise (7). Mn and Cd were absent in the dust fall on all buildings. As pollution occurred only in the dust fall on the low-rise buildings. Slight Cr, Ni, Cu, and Pb pollution, as well as moderate Zn pollution, were detected in the dust fall on the low-rise buildings. In the dust fall on medium-rise buildings, there was slight Cr, Ni, Pb, Cu, and Zn pollution. In the dust fall on high-rise buildings, there was slight Cr, Ni, Pb, and Cu pollution, but moderate Zn pollution. Zn was the major heavy metal pollutant in the dust fall on all buildings.

#### 3.3.2. Potential Ecological Risk Index

Figure 4 shows the potential ecological risk assessments for the heavy metals in the dust fall on the buildings differing in height. The average potential ecological risk indices for the dust fall on the low-, medium-, and high-rise buildings were in the orders of Cd > Cu > As > Ni > Cr > Pb > Zn > Mn; Cd > Cu > As > Pb > Ni > Cr > Zn > Mn; and Cd > Cu > As > Pb > Zn > Ni > Cr > Mn, respectively. The ecological risk of Cd was very high for all three types of buildings. The single-factor ecological hazard ratings for the other seven heavy metals were all <40. Hence, their ecological risks were low. The contribution rate of Cd was the highest (72%) and far exceeded those of all other heavy metals. Thus, Cd had a very high pollution level. The total potential ecological risk hazard indices (RIs) were in the order of low- (337.02) > high- (286.90) > medium- (284.40) rise buildings. The total potential ecological RIs and the ecological hazard ratings for the heavy metals in the dust fall on the low-rise buildings were significantly higher than those for the heavy metals in the dust fall on medium- and high-rise buildings. The RIs for medium- and high-rise buildings were <300, and their ecological risk assessments were moderate. The foregoing results confirm heavy metal, and especially Cd, pollution in the atmospheric dust fall of Beijing.

### 3.4. Analyses of Dust Fall and Heavy Metal Sources

#### 3.4.1. Correlation Analyses

Correlation analyses can indicate whether heavy metals have the same sources. The Pearson’s correlations among heavy metals were calculated using SPSS (Table 6). Cr and Ni were highly significantly correlated (*p* < 0.01; r = 0.95). The correlation coefficients between Cu and Mn, Cu and Zn, Cu and Cd, and Cu and Pb were 0.639, 0.734, 0.594, and 0.665, respectively. As they were all >0.5, these heavy metals were all strongly correlated. The correlation coefficients between Zn and Cd, and Zn and Pb were 0.671 and 0.615, respectively. The coefficient of correlation between Cd and Pb was 0.593. The coefficients of correlation between Mn and Zn, Ni and Pb, and As and Cd were 0.426, 0.389, and 0.394, respectively. Hence, there were significant positive correlations among heavy metals (*p* < 0.05). The foregoing elements might have originated from the same source and were not correlated with any other elements. The strong correlations among Cd, Pb, and Zn suggest that the levels of these heavy metals are substantially influenced by human activity.

#### 3.4.2. Principal Component Analysis

To determine the dust reduction sources, PCA was conducted on the eight heavy metals according to the results of the preceding correlation analysis (Table 7). The first three principal components were selected in the order of large to small eigenvalues. The total contribution of the three principal components was 80.6%. Thus, they reliably represented the analytical results for all data.

Table 8 shows that the first principal component included mainly Cr, Cu, Zn, Cd, and Pb, and the contribution rate was 49.08%. Pb was formerly associated primarily with automobile exhaust pollution, Pb in automobile exhaust comes from gasoline [32]. Cr, Zn, and Cu originate from the wear of tires and other automobile parts [33]. Cd is derived from automobile fuel [34,35]. The sampling point was near the main artery of the Third Ring Road in Beijing. It is reasonable to assume that traffic factors had a significant impact here. For this reason, component 1 was the traffic source. The contribution rate of component 2 was 20.75%, and Cr and Ni had the highest loads. Table 6 shows a strong correlation between these metals (0.950). Hence, they probably originated from the same source. Thus far, it is the parent material that mainly determines the Cr and Ni content in Beijing soil [36]. Consequently, component 2 is a natural source. Component 3 had the highest As load. This element abounds in coal-fired emissions [37]. Hence, component 3 is an industrial source.

#### 3.4.3. Rare Earth Element Analysis

Figure 5 consists of box plots indicating the REE contents in the dust fall on buildings differing in height. The average ΣREEs in the dust fall on the low-, medium-, and high-rise buildings were 88.63 mg·kg^−1^, 111.86 mg·kg^−1^, and 117.39 mg·kg^−1^, respectively. Thus, the REE contents increased with building height. The opposite trend was true for the heavy metal elements.

The total REE concentrations in the dust fall on the low-, medium-, and high-rise buildings were in the order of Ce > La > Nd > Pr > Sm > Gd > Dy > Er > Yb > Eu > Ho > Tb > Tm > Lu. This sequence is essentially consistent with that for REEs in the continental crust [38,39]. Therefore, the REEs in dust fall partially originate from natural sources. This finding was consistent with the results for component 2 of the PCA. Ce and La accounted for the largest proportions (50% and 18%, respectively) of all REEs. Both metals are widely used in catalytic converters [40]. For this reason, certain REEs originate from traffic sources. The sampling point was located in urban Beijing which has a voluminous traffic flow near the main roads. The analyses of REE dust sources were consistent with those for component 1 of the PCA.

## 4. Discussion

The geoaccumulation index mainly considers the enrichment degree of exogenous heavy metals. On this basis, the potential ecological risk assessment accounts for the toxic biological effects of different toxic heavy metals. This is the main reason for the difference between the two evaluation results. The research results show that the pollution degree of low-rise buildings is more serious, which may be due to the influence of wind. The size of wind speed is directly proportional to the movement degree of particles. The higher the wind speed, the larger the displacement of dust particles in the atmosphere, and the size of particles will affect the suspension height, making it easier for low-rise buildings to collect particles with different particle sizes, so that the samples collected from low-rise buildings are seriously polluted. Therefore, during dust reduction in the study area, the placement height of the dust collecting cylinder at the sampling point should be kept, as much as possible, at the same height to reduce the error caused by the experiment design. Comparing the research of Xiong Qiulin et al. on Beijing’s winter dust fall, the content of heavy metals measured in this study is low, which may be due to the seasonality of dust fall. The content of heavy metals in particulate matter varies in different seasons. Hu Jing et al. [41] analyzed the pollution characteristics of heavy metals in atmospheric PM2.5 in autumn and winter in Guiyang City, and found that the concentrations of Cr, As, Mn, Cu, and Pb in winter were significantly higher than those in autumn. From the above research it can be shown that the temporal distribution characteristics of heavy metals in atmospheric particles are different in different seasons, and generally show the characteristic that distribution in winter is larger than in the other three seasons. This may be due to the strong atmospheric stability in winter, the difficulty in diffusion of particles, and the need for heating in winter, which increases coal combustion and leads to the increase in characteristic elements of coal-fired emissions. The sampling time of this study is one year, and it is normal that the average content of heavy metals in dust fall is lower than that in winter, which is consistent with the previous research results.

The focus of this study was the correlation between the hazards and sources of dust fall on the tops of buildings and the building height. We had only 14 sampling points for the research, and extreme weather greatly affected some results. The results of the analysis of data collected during months of extreme weather (such as sandstorms) were significantly altered relative to the data from months with normal weather. We could curtail the impact of extreme weather on the overall data by extending the sampling time and collecting more samples. The sampling points were concentrated in a relatively small area, which may not represent the correlation between buildings and dust reduction across the whole of Beijing. Future research will stem from the present results, increasing the number of sampling points and expanding the sampling area, rendering the research results more representative.

## 5. Conclusions

(1) The average Mn content in the dust fall on medium-rise buildings was slightly lower than the background Mn levels in Beijing soil. The concentrations of Cr, Ni, Cu, Zn, As, Cd, and Pb in the dust fall on all buildings exceeded the respective background values in Beijing soil.

(2) The *EF* evaluation showed that both Zn and Cd were significantly enriched in the dust fall on all buildings. In contrast, Cu and Pb were only moderately enriched. The combination of the *EF* evaluation and the PCA revealed that traffic sources contributed 48.08% of the dust fall heavy metal burden in the study area.

(3) The order of the degree of heavy metal pollution in the dust fall was low- > high- > medium-rise buildings. Mn and Cd were pollution-free in all areas, whereas As presented a low pollution risk. Cr, Ni, Cu, and Pb were either at a slight pollution level or posed a mild contamination risk in the dust fall on all buildings. Zn had the highest pollution risk in the dust fall on medium-rise buildings but reached a moderate pollution level in the dust fall on low- and high-rise buildings.

(4) The potential ecological risk assessment indicated that Cd was associated with high ecological risk in the dust fall on all buildings. Hence, Cd contamination should be strictly regulated. The RIs for the low-, medium-, and high-rise buildings were 337.02, 284.40, and 286.90, respectively. Low-rise buildings were at high ecological risk, whereas medium- and high-rise buildings were at moderate ecological risk.

(5) The sources of heavy metals in the dust fall can be divided into three categories. In the first, all eight heavy metals were derived from traffic sources. In the second, Cr and Ni originated from natural sources. In the third, all heavy metals were of industrial origin. According to the PCA, the sources of REE in the dust fall were essentially consistent with those for the heavy metals.

(6) The evaluation results show that the height of buildings impacts the enrichment of heavy metals in the dust fall. These findings provide valuable information for determining the height of buildings and the pollution sources of atmospheric dust fall and may help to design improved control strategies. Nonetheless, the sampling range of this study was small, and the research results were not universal; they can only provide some theoretical support for dust control. In the future, multi-point sampling strategies in different areas of Beijing should be implemented to explore the spatial differences in dust pollution related to building height in Beijing.

## Figures and Tables

**Figure 1 ijerph-19-11376-f001:**
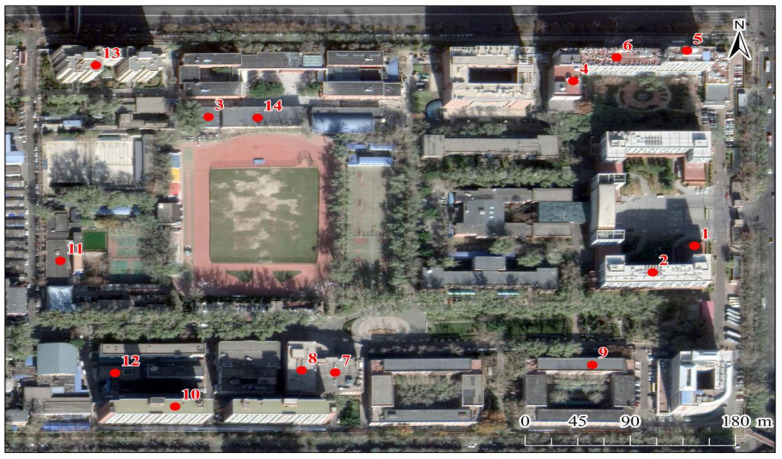
Distribution of the atmospheric dust fall sampling points in the study area.

**Figure 2 ijerph-19-11376-f002:**
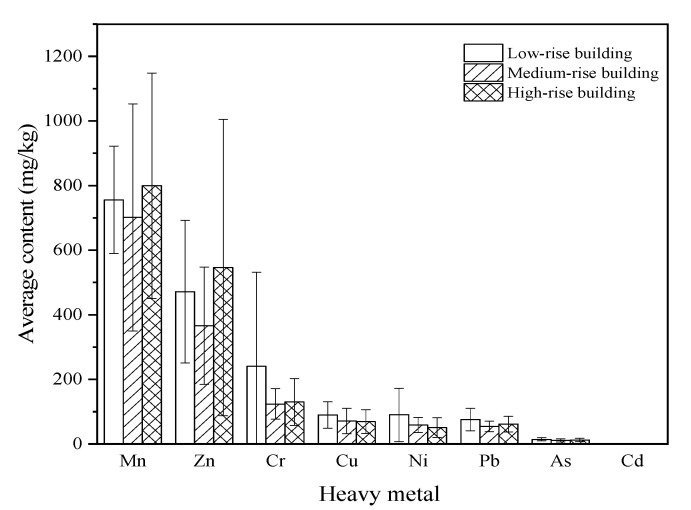
Variations in the average heavy metal content in the dust fall on low-, medium-, and high-rise buildings.

**Figure 3 ijerph-19-11376-f003:**
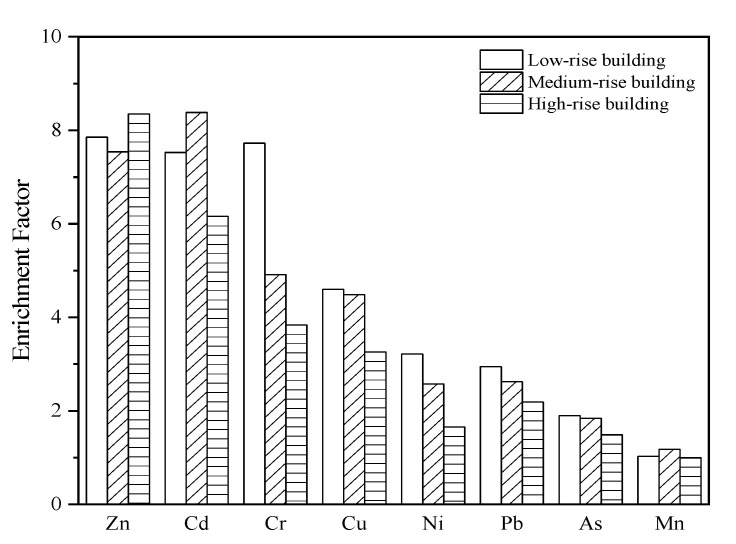
Enrichment factors for heavy metals in the dust fall on buildings differing in height.

**Figure 4 ijerph-19-11376-f004:**
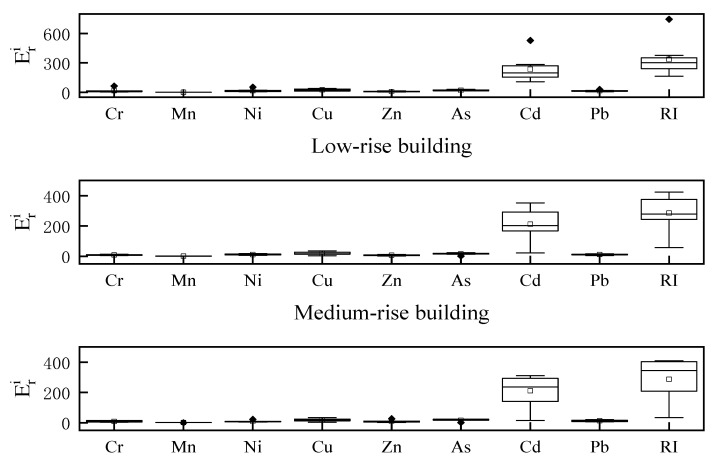
Potential ecological risk indices of atmospheric dust fall on buildings differing in height. RI: The total potential ecological risk hazard indices.

**Figure 5 ijerph-19-11376-f005:**
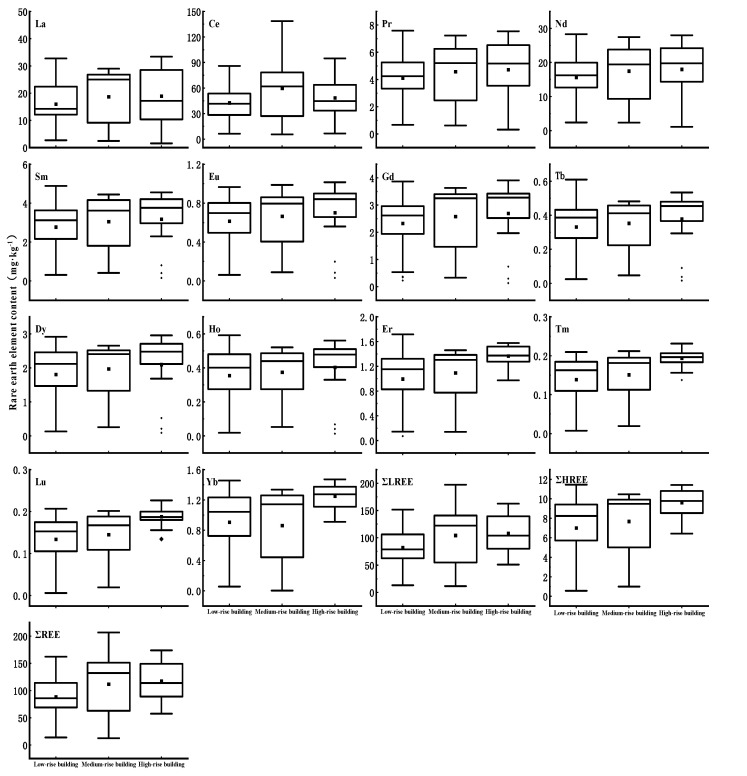
Box plot of rare earth element contents in atmospheric dust fall on buildings differing in height.

**Table 1 ijerph-19-11376-t001:** *EF* classification criteria [15].

Pollution Level	*EF*	Enrichment Degree
I	*EF* < 1	No enrichment
	1 ≤ *EF* < 2	Slight enrichment
II	2 ≤ *EF* < 5	Moderate enrichment
III	5 ≤ *EF* < 20	Significant enrichment
IV	20 ≤ *EF* < 40	Strong enrichment
V	*EF* ≥ 40	Extremely strong enrichment

**Table 2 ijerph-19-11376-t002:** Indices of geoaccumulation and degrees of pollution.

Grade	*I* _geo_	Degree of Pollution
0	*I*_geo_ ≤ 0	Pollution-free
1	0 < *I*_geo_ ≤ 1	Slight pollution
2	1 < *I*_geo_ ≤ 2	Mild pollution
3	2 < *I*_geo_ ≤ 3	Moderate pollution
4	3 < *I*_geo_ ≤ 4	High pollution
5	4 < *I*_geo_ ≤ 5	Heavy pollution
6	*I*_geo_ > 5	Severe pollution

**Table 3 ijerph-19-11376-t003:** Potential ecological risk assessment indices and their classifications.

$E_{r}^{i}$ (Single-Factor Potential Ecological Hazard Coefficient)	*RI* (Total Potential Ecological Risk Hazard Index)	Potential Ecological Risk Degree
<40	<150	Low ecological risk
40–80	150–300	Moderate ecological risk
80–160	300–600	Considerable ecological risk
160–320	600–1200	High ecological risk
≥320	≥1200	Very high ecological risk

**Table 4 ijerph-19-11376-t004:** Statistics of heavy metal elements in dust fall on various buildings.

Heavy Metal	Data	Low-Rise Building	Medium-Rise Building	High-Rise Building	Background Value (mg/kg)
Cr	Average	265.1	123.7	130.1	29.8
	Standard deviation (SD)	291.7	47.3	72.4	
	Coefficient of variation (CV)	1.2	0.4	0.6	
Mn	Average	756.0	701.4	799.5	705
	SD	166.3	351.5	348.9	
	CV	0.2	0.5	0.4	
Ni	Average	89.1	58.3	50.5	26.8
	SD	82.6	23.3	30.3	
	CV	0.9	0.4	0.6	
Cu	Average	89.8	70.8	69.3	18.7
	SD	41.3	39.5	36.8	
	CV	0.5	0.6	0.5	
Zn	Average	471.3	366.1	546.6	57.5
	SD	220.8	181.8	458.9	
	CV	0.5	0.5	0.8	
As	Average	14.0	11.0	12.0	7.09
	SD	4.9	4.8	5.3	
	CV	0.3	0.4	0.4	
Cd	Average	0.9	0.8	0.8	0.119
	SD	0.5	0.4	0.4	
	CV	0.6	0.5	0.5	
Pb	Average	75.7	54.5	61.2	24.6
	SD	34.7	16.6	24.1	
	CV	0.5	0.3	0.4	

The background value is the background heavy metal content in Beijing soil.

**Table 5 ijerph-19-11376-t005:** Geoaccumulation indices for dust fall on buildings of different heights.

Heavy Metal	Data	Low-Rise Building	Medium-Rise Building	High-Rise Building
Cr	*I* _geo_	1.35	0.39	0.46
	Pollution degree	Mild pollution	Slight pollution	Slight pollution
	Pollution level	2	1	1
Mn	*I* _geo_	−0.13	−0.24	−0.05
	Pollution degree	Pollution-free	Pollution-free	Pollution-free
	Pollution level	0	0	0
Ni	*I* _geo_	1.21	0.58	0.37
	Pollution degree	Mild pollution	Slight pollution	Slight pollution
	Pollution level	2	1	1
Cu	*I* _geo_	1.38	1.04	1.01
	Pollution degree	Mild pollution	Mild pollution	Mild pollution
	Pollution level	2	2	2
Zn	*I* _geo_	2.23	1.86	2.44
	Pollution degree	Moderate pollution	Mild pollution	Moderate pollution
	Pollution level	3	2	3
As	*I* _geo_	0.04	−0.31	−0.18
	Pollution degree	Slight pollution	Pollution-free	Pollution-free
	Pollution level	2	0	0
Cd	*I* _geo_	−7.74	−7.89	−7.91
	Pollution degree	Pollution-free	Pollution-free	Pollution-free
	Pollution level	0	0	0
Pb	*I* _geo_	1.01	0.54	0.71
	Pollution degree	Mild pollution	Slight pollution	Slight pollution
	Pollution level	2	1	1

**Table 6 ijerph-19-11376-t006:** Coefficients of correlation between heavy metals in dust fall.

	Cr	Mn	Ni	Cu	Zn	As	Cd	Pb
Cr	1							
Mn	0.052	1						
Ni	0.950 **	−0.052	1					
Cu	0.362	0.639 **	0.252	1				
Zn	0.370	0.426 *	0.217	0.734 **	1			
As	0.349	0.341	0.241	0.346	0.287	1		
Cd	0.339	0.386	0.257	0.594 **	0.671 **	0.394 *	1	
Pb	0.333	0.329	0.389 *	0.665 **	0.615 **	0.247	0.593 **	1

* *p* < 0.05; ** *p* < 0.01.

**Table 7 ijerph-19-11376-t007:** Principal component analysis.

	Initial Eigenvalue			Sum of Squares of Loads		
	Total	Variance (%)	Cumulative (%)	Total	Variance (%)	Cumulative (%)
1	3.926	49.077	49.077	3.926	49.077	49.077
2	1.660	20.745	69.822	1.660	20.745	69.822
3	0.870	10.875	80.697	0.870	10.875	80.697
4	0.602	7.523	88.220			
5	0.416	5.203	93.423			
6	0.332	4.155	97.578			
7	0.180	2.255	99.833			
8	0.013	0.167	100.000			

**Table 8 ijerph-19-11376-t008:** Heavy metal factor load matrix for dust reduction.

Heavy Metal	Component		
	1	2	3
Cr	0.640	**0.729**	0.070
Mn	0.560	−0.551	0.294
Ni	0.548	**0.811**	−0.028
Cu	**0.853**	−0.285	−0.084
Zn	**0.813**	−0.226	−0.241
As	0.539	0.022	**0.763**
Cd	**0.786**	−0.169	−0.081
Pb	**0.778**	−0.070	−0.352

Boldface font indicates a higher value of each component load.

## Data Availability

Not applicable.

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
