# Peer review of "Heavy Metal Pollution Characteristics and Source Analysis in the Dust Fall on Buildings of Different Heights"

_ijerph, 2022, doi:10.3390/ijerph191811376_

Round 1
Reviewer 1 Report
In this study, eight heavy metal components were sampled from the dust falling at sample points, and EF, Igeo, PCA and other methods were used to analyze the characteristics and sources of heavy metal pollution in buildings. On the whole, the idea of the research is clear, but the research innovation is insufficient. Part of the logic of the research process is not clear, and the research results have no corresponding basis to prove. There are also some problems in the format and content of the paper. Therefore, I believe that this manuscript needs to be significantly revised before publication. My specific comments are as follows:
(1) Only 14 sample points were selected in the study, with a very small coverage. Then why can this area represent Beijing? And the sample point is in the school area, can it represent the buildings under general conditions? Can we get general conclusions about air pollution in Beijing with such research data? If yes, please explain the reasons clearly. If not, please explain the significance of the research and the application scope of the conclusions.
(2) The paper lacks the introduction of the characteristics of the research area and the surrounding environment, so the analysis basis of pollution sources in the following paper is weak.
(3) This study only introduces the distribution of sample points, but does not mention why sample points are arranged in this way, and what is the selection logic of sample points for high school low-rise buildings. At the same time, it is noted that some sample points are nearly adjacent, not scattered, and the randomness of sample points is not strong.
(4) The paper did not mention the definition standards of the selected low, medium and high-rise buildings. In addition, for this research question, what basis and significance are there for the research perspective to be cut into the low, medium and high-rise buildings?
(5) In the source analysis, the content of relevant elements is used to determine the source. Please supplement the relevant literature basis in the paper.
(6) There is no Bi and Bn in formula (1), please correct the interpretation of the two elements in this formula.
(7) In the last paragraph of section 7.2.3, please add relevant literature on the basis of reference element selection. If possible, you can compare different reference element selection methods to introduce the rationality of this reference element selection method in more detail. In addition, please explain the reason why iron element is selected as the only reference element among the four elements.
(8) Please supplement the application of principal component analysis in this study in more detail in Section 2.5.1.
(9) Table 3 is not mentioned in the article, please briefly state it in the appropriate place.
(10) The ninth reference in the introduction does not mention principal component analysis (PCA), please correct the reference; The serial number in FIG. 1 is clearly marked; The position of serial number of formula (1) - (4) should be unified; The table described at the beginning of paragraph 2 in Section 3.1 should be Table 4; Please adjust the center form of the tables and figures in the article.
Author Response
(1) Only 14 sample points were selected in the study, with a very small coverage. Then why can this area represent Beijing? And the sample point is in the school area, can it represent the buildings under general conditions? Can we get general conclusions about air pollution in Beijing with such research data? If yes, please explain the reasons clearly. If not, please explain the significance of the research and the application scope of the conclusions.
Response 1: The evaluation results show that the height of buildings impacts the enrichment of heavy metals in the dust fall. These findings provide valuable information for determining the height of buildings and the pollution sources of atmospheric dust fall, and may help to design improved control strategies. Nonetheless, the sampling range of this study was small, and the research results were not universal; they can only provide some theoretical support for dust control. In the future, multi-point sampling strategies in different areas of Beijing should be implemented to explore the spatial differences in dust pollution related to building height in Beijing.
(2) The paper lacks the introduction of the characteristics of the research area and the surrounding environment, so the analysis basis of pollution sources in the following paper is weak.
Response 2: A basic introduction to the surrounding environment has been added to the study area.
(3) This study only introduces the distribution of sample points, but does not mention why sample points are arranged in this way, and what is the selection logic of sample points for high school low-rise buildings. At the same time, it is noted that some sample points are nearly adjacent, not scattered, and the randomness of sample points is not strong.
Response 3: The distribution methods and principles have been added to the study area and sample collection.The sampling points are evenly distributed, including five, four, and five on low-, medium-, and high-rise buildings. We attempted to ensure that adjacent sampling points avoided building height zoning of the same type to reduce the impact of randomness.
(4) The paper did not mention the definition standards of the selected low, medium and high-rise buildings. In addition, for this research question, what basis and significance are there for the research perspective to be cut into the low, medium and high-rise buildings?
Response 4: A specific introduction to the classification of building height has been added in the preface.These buildings not only block the flow of air—their top surfaces also become the main place for dust accumulation, constituting an important secondary dust source.The building density is steadily increasing in urban Beijing. However, relatively little research has been conducted on dust fall reduction and re-entrainment on building rooftops. Hence, we examined the properties of atmospheric dust fall on buildings differing in height, the regional atmospheric pollution status, and meteorological conditions.
(5) In the source analysis, the content of relevant elements is used to determine the source. Please supplement the relevant literature basis in the paper.
Response 5: Two references 37-38 have been added to supplement the source analysis.
(6) There is no Bi and Bn in formula (1), please correct the interpretation of the two elements in this formula.
Response 6: Formula 1 has been corrected.
(7) In the last paragraph of section 1.2.3, please add relevant literature on the basis of reference element selection. If possible, you can compare different reference element selection methods to introduce the rationality of this reference element selection method in more detail. In addition, please explain the reason why iron element is selected as the only reference element among the four elements.
Response 7: According to the method of reference element selection reported by Fan Xiaoting et al., elements with natural sources, stable chemical properties, and high content in the crust were selected . Here, Fe was selected as the reference element.
(8) Please supplement the application of principal component analysis in this study in more detail in Section 2.5.1.
Response 8: The specific application of principal component analysis in this study has been added in 2.5.1
(9) Table 3 is not mentioned in the article, please briefly state it in the appropriate place.
Response 9: Already mentioned in the second paragraph of 2.4.2.
(10) The ninth reference in the introduction does not mention principal component analysis (PCA), please correct the reference; The serial number in FIG. 1 is clearly marked; The position of serial number of formula (1) - (4) should be unified; The table described at the beginning of paragraph 2 in Section 3.1 should be Table 4; Please adjust the center form of the tables and figures in the article.
Response 10: The ninth reference has been corrected. The sampling point mark in Figure 1 has been enlarged and more eye-catching. The positions of formulas (1) - (4) have been changed to right alignment. The beginning of paragraph 2 of 3.1 has been changed to table 4. The graph and table center have been adjusted.

Reviewer 2 Report
The article includes data from a year of analysis of samples collected monthly at 14 different stations. I consider the methodology appropriated. I have a few comments:
- - Figure 1: needs to be improved, I can’t see the sampling stations in my printed copy and it’s not very good quality on the PDF.
- - For section 2.3: Why not using another element like Al and Ti for reference to calculate EF? Their solubility is lower and it’s been described that Fe particles have increased in the Chinese atmosphere due to the increase of Fe and steel industries (Zhou et al., 2020, PLOS ONE). If this is happening, Fe concentrations might be affected. This change might affect the results in section 3.2.
- - Table 4: Can you add some more info about what Background values are? You introduced this in section 2.4 but it will be good to mention in the legend again.
- - Figure 4: please indicate what RI means (it’s in the text but it will help when you are checking the figure)
- - Section 3.4.2: second paragraph, after mentioning Pb origins, you need to add a reference, the same for the next sentence. The authors mentioned that Pb is from the automobile exhaust pollution, is this coming from the fuel? Or the material?
- - Section 3.4.3: When discussing the REE, the authors should consider working on the second paragraph, also including REE in the PCA. You can’t compare results with PCA results if REE values are not included. If those data exist, the authors should include a table with them, but also a PCA that includes heavy metals and REE.
- - Figure 5: some text is still in Chinese
- - In discussion, the authors mentioned how extreme weather affected the samples. This should be better explained in the method section when talking about the sampling site or in a new section in results where you can see the metal concentrations and meteorological conditions.
- - Some English corrections are required
Author Response
1. Figure 1: needs to be improved, I can’t see the sampling stations in my printed copy and it’s not very good quality on the PDF.
Response 1: The sampling points in Figure 1 have been processed to make them more obvious.
2. For section 2.3: Why not using another element like Al and Ti for reference to calculate EF? Their solubility is lower and it’s been described that Fe particles have increased in the Chinese atmosphere due to the increase of Fe and steel industries (Zhou et al., 2020, PLOS ONE). If this is happening, Fe concentrations might be affected. This change might affect the results in section 3.2.
Response 2: According to the method of reference element selection reported by Fan Xiaoting et al., elements with natural sources, stable chemical properties, and high content in the crust were selected .Here, Fe was selected as the reference element.
3. Table 4: Can you add some more info about what Background values are? You introduced this in section 2.4 but it will be good to mention in the legend again.
Response 3: 2.4 the background value is the geochemical background of heavy metal element, Table 4 is the background heavy metal content in Beijing soil,It is introduced in the second paragraph of Article 3.1 that the background heavy metal content in Beijing soil in Beijing is selected according to Literature 29-30.It has been noted and distinguished below the table.
4. Figure 4: please indicate what RI means (it’s in the text but it will help when you are checking the figure)
Response 4: The meaning of RI has been added below figure 4.
5. Section 3.4.2: second paragraph, after mentioning Pb origins, you need to add a reference, the same for the next sentence. The authors mentioned that Pb is from the automobile exhaust pollution, is this coming from the fuel? Or the material?
Response 5: References have been added in 3.4.2 and the source of Pb has been explained.
6. Section 3.4.3: When discussing the REE, the authors should consider working on the second paragraph, also including REE in the PCA. You can’t compare results with PCA results if REE values are not included. If those data exist, the authors should include a table with them, but also a PCA that includes heavy metals and REE.
Response 6:As a group of elements with special geochemical properties, REE's composition and distribution pattern will not be affected by weathering, transportation, sedimentation and diagenesis, and there is less obvious chemical separation in the environmental process. The source of dustfall can be determined by using the content of rare earth elements in Dustfall and some important rare earth element parameters. This paper only compares the results of two source analysis. PCA infers the source of dustfall through the analysis of heavy metals, and rare earth elements infer the source of dustfall through the analysis of content. In this paper, two different methods are used to verify whether the results of the two methods are consistent.
7. Figure 5: some text is still in Chinese
Response 7: The Chinese in the figure has been changed.
8. In discussion, the authors mentioned how extreme weather affected the samples. This should be better explained in the method section when talking about the sampling site or in a new section in results where you can see the metal concentrations and meteorological conditions.
Response 8: The impact of extreme weather has been reduced by increasing the sampling data time, and the Countermeasures of extreme weather impact samples are added to the method。

Round 2
Reviewer 1 Report
Thank you for your reply to the questions raised last time. The content and structure of this article have been supplemented and improved to some extent. However, I think there are some minor problems that need to be further explained. The details are as follows:
1. The second sentence of the abstract is slightly inconsistent with the article, and the sample points do not represent Beijing. Please check it.
2. This paper explores the dust fall characteristics of buildings with different heights. If there are arrays of adjacent buildings in the sample points, will the adjacency relationship of buildings with different heights affect the dust fall characteristics? When the influence of adjacent buildings is not considered, should we avoid selecting adjacent buildings when selecting sample points. Please give your opinions.
3. Although the introduction of the article gives the definition of different heights of buildings, it is not obvious that the height of buildings is different from some sample points in the figure. Please give a brief description of the specific height of these buildings in the text when dividing the sample into three types of buildings.
4. After the last revision, the description of the surrounding environment in the article specifically mentions that there are no industrial pollution sources nearby, but when the pollution sources are explored later, it is said that heavy metals are from industrial pollution sources. Please give an appropriate explanation to this contradiction to make itself consistent.
Author Response
(1) The second sentence of the abstract is slightly inconsistent with the article, and the sample points do not represent Beijing. Please check it.
Response 1: The second paragraph of the abstract has been revised accordingly.
(2) This paper explores the dust fall characteristics of buildings with different heights. If there are arrays of adjacent buildings in the sample points, will the adjacency relationship of buildings with different heights affect the dust fall characteristics? When the influence of adjacent buildings is not considered, should we avoid selecting adjacent buildings when selecting sample points. Please give your opinions.
Response 2: A basic introduction to the surrounding environment has been added to the study area.
(3) Although the introduction of the article gives the definition of different heights of buildings, it is not obvious that the height of buildings is different from some sample points in the figure. Please give a brief description of the specific height of these buildings in the text when dividing the sample into three types of buildings.
Response 3: The specific height of buildings at each point has been added in 2.1 study area and sample collection.
(4) After the last revision, the description of the surrounding environment in the article specifically mentions that there are no industrial pollution sources nearby, but when the pollution sources are explored later, it is said that heavy metals are from industrial pollution sources. Please give an appropriate explanation to this contradiction to make itself consistent.
Response 4: The dust will enter the collection area through natural sedimentation, precipitation or wind force, and the transportation distance will change with the size of particles, thus affecting the assessment of pollution sources in the collection area. The surrounding environment of the sampling point is described in the paper. Compared with the dust source, the scope is still very small.

Reviewer 2 Report
I can see the authors improved the article and correct errors from the previuos version. I consider this article is good for publishing.
Author Response
"Please see the attachment."
